# Building Vaccine Readiness for Future Pandemics: Insights from COVID-19 Vaccine Intent and Uptake

**DOI:** 10.3390/vaccines13121201

**Published:** 2025-11-28

**Authors:** Jeanine P. D. Guidry, Linnea I. Laestadius, Carrie A. Miller, Michael P. Stevens, Candace W. Burton, Kellie E. Carlyle, Paul B. Perrin

**Affiliations:** 1Department of Communication and Cognition, Tilburg University, 5045 LJ Tilburg, The Netherlands; 2Joseph J. Zilber College of Public Health, University of Wisconsin-Milwaukee, Milwaukee, WI 53205, USA; llaestad@uwm.edu; 3Department of Family Medicine and Population Health, Virginia Commonwealth University School of Medicine, Richmond, VA 23298, USA; carrie.a.miller@vcuhealth.org; 4Department of Internal Medicine, Division of Infectious Diseases, Morgantown, WV 26506, USA; mike.stevens@wvumedicine.org; 5School of Nursing, York University, Toronto, ON M3J 1P3, Canada; cwburton@yorku.ca; 6Department of Social and Behavioral Sciences, Virginia Commonwealth University School of Public Health, Richmond, VA 23219, USA; kecarlyle@vcu.edu; 7School of Data Science, Department of Psychology, University of Virginia, Charlottesville, VA 22903, USA; perrin@virginia.edu

**Keywords:** COVID-19 vaccine, health belief model, pandemic preparedness, trust in government

## Abstract

Background/Objectives: This longitudinal study investigated psychosocial predictors of COVID-19 vaccine intentions before vaccine availability (July 2020) and vaccine uptake or ongoing intent after widespread vaccine rollout (April 2021) using constructs from the Health Belief Model (HBM) and a measure of trust in government. Methods: A U.S. adult sample (N = 142) completed surveys at two time points: prior to and following the release of COVID-19 vaccines. Key predictors included demographics, trust in government, and HBM constructs. Hierarchical logistic regression was used to predict vaccine intent and uptake at both time points. Results: At Time 1, intent to vaccinate was significantly predicted by higher perceived susceptibility (*p* = 0.038), greater perceived benefits (*p* < 0.001), and lower perceived barriers (*p* = 0.002). Trust in government was not a significant predictor. At Time 2, vaccine uptake/ongoing intent was significantly predicted by higher trust in government (*p* = 0.047), greater perceived benefits (*p* < 0.001), and lower perceived barriers (*p* = 0.002). Perceived susceptibility was no longer a significant predictor. Between time points, trust in government and self-efficacy increased, while perceived severity and barriers decreased. Conclusions: Perceived benefits and barriers were robust predictors of vaccine behavior across both time points. Trust in government became a stronger predictor once vaccines were available, underscoring the importance of building and maintaining public trust throughout a health crisis. Messaging should emphasize vaccine benefits, proactively address barriers, and adapt over time as public perceptions shift. These findings inform strategies for enhancing vaccine confidence and readiness in future pandemics.

## 1. Introduction

While the World Health Organization’s (WHO) public health emergency declaration for COVID-19 is behind us, there is broad scientific and public health consensus that future pandemics are inevitable [1,2]. Preparing for these future outbreaks, whether caused by another coronavirus, avian influenza A, or an entirely different pathogen requires critical reflection on the successes and failures of the COVID-19 response [3]. Since before the COVID-19 pandemic, the WHO has encouraged member nations to develop a pandemic preparedness plan, of which vaccinations are a cornerstone [4]. Insights into vaccine-related beliefs, intentions, and the most robust predictors of vaccine uptake can inform proactive infectious disease management and health communication strategies before the next outbreak [3].

### 1.1. COVID-19 Vaccine Development and Uptake

In early 2020, soon after the World Health Organization (WHO) declared the COVID-19 outbreak a pandemic [5], researchers leveraged decades of prior work on mRNA and viral-vector technologies to initiate COVID-19 vaccine development at unprecedented speed. Emergency use authorizations (EUAs) for both Pfizer–BioNTech and Moderna vaccines were issued in December 2020 [6]. The first full U.S. FDA approval came in August 2021 for Pfizer–BioNTech’s (ages 16+), followed by Moderna in January 2022 (ages 18+) [6]. These approvals were based on six months or more of safety and efficacy data following their EUAs. Uptake during the first year, once the COVID-19 vaccines became available, was generally higher than the pre-release studies on intent predicted [7,8,9]. In a 2022 survey with 19,502 respondents in Texas, 78% reported having been vaccinated against COVID-19, while 22% did not [10]. A population-based study of the Netherlands concluded that 84% of all eligible adults had received at least one dose of the COVID-19 vaccine by September 2022 [11]. The CDC reported that by 4 July 2021, 67% of U.S. adults had received at least one dose of a COVID-19 vaccine [12].

A longitudinal study of U.S. adults (N = 7420) reported an overall decrease in COVID-19 vaccine hesitancy from 46.0% in October 2020 to 35.2% in March 2021 [13]. Despite these initially encouraging trends, however, the pandemic also heightened public concerns about vaccinations, with routine vaccination coverage dropping and parental vaccine hesitancy increasing [3,4,5]. Research found that 23.1% of respondents across 23 countries reported being less willing to receive non-COVID-19 vaccines due to their pandemic-era experiences [14], raising serious concerns about the broader and lasting impacts of the pandemic on vaccine confidence.

### 1.2. Theoretical Explanations for Vaccine Behavior

Psychosocial predictors of vaccination behavior have been well-studied through the lens of health behavior theories and help contextualize these patterns in vaccine intent and uptake. According to Trust Determination Theory, trust is the most powerful influence on how people make risk-related decisions. The more trusted the information source, the more acceptable the messages, messengers, and channels for acquiring information [15]. Trust in authorities involves evaluating leaders based on their competence, fairness, honesty, care, accountability, and transparency [15,16]. Prior to the COVID-19 pandemic, the U.S., like many other countries, had experienced a decades-long decline in trust in government, which can significantly influence public receptivity to health guidance during crises [17]. Recent studies confirm this, with trust in government serving as a critical determinant of public confidence in COVID-19 vaccination efforts during the pandemic, and distrust in government-provided information being associated with COVID-19 vaccine hesitancy and lower COVID-19 vaccine uptake [18,19,20,21]. This underscores the importance of considering government trust in models of vaccine intent and uptake.

In addition to trust, the Health Belief Model (HBM) [22,23,24,25] argues that the likelihood of someone adopting a specific health behavior is determined by belief in a personal threat of illness or disease, together with belief in the effectiveness of the recommended behavior (see Figure 1). Its constructs, as applied to COVID-19 and the COVID-19 vaccine, are perceived severity of and perceived susceptibility to COVID-19, perceived benefits of and barriers to getting a COVID-19 vaccine, self-efficacy to overcome COVID-19 vaccination barriers, and cues to action to get the vaccine [26,27,28]. The HBM has been used extensively to explain COVID-19 vaccine uptake and intent [26,27,28,29].

### 1.3. Study Hypotheses

Using trust in government and the HBM as frameworks, the aim of this study was to assess psychosocial predictors of U.S. adults’ willingness in July 2020 to get a then-future COVID-19 vaccine and both whether and how these predictors changed in April 2021, after the vaccines were broadly available to the public. Based on the previous literature and theory, the following hypotheses were proposed:

**Hypothesis** **1:**
*Perceived benefits of the COVID-19 vaccine and perceived barriers to vaccination will be stronger predictors of vaccine uptake/intent after vaccine rollout (Time 2) than before vaccine availability (Time 1). Prior studies suggest that while perceived susceptibility and severity may initially drive vaccine intent, once a vaccine becomes available, the salience of its perceived benefits and barriers increases in influencing behavior [30,31,32]. This reflects a shift from abstract risk appraisal to practical decision-making.*


**Hypothesis** **2:**
*Trust in government will be a stronger predictor of vaccine uptake/intent at Time 2 than at Time 1. According to Trust Determination Theory, trust becomes more influential as public institutions begin to implement health policies [33]. In Time 1, when vaccines were still hypothetical, trust may have had less direct relevance. At Time 2, however, trust likely influenced acceptance of government-led vaccination efforts and communications.*


**Hypothesis** **3:**
*Perceived severity of COVID-19 will decrease from Time 1 to Time 2, while self-efficacy for vaccination and trust in government will increase. This hypothesis stems from public shifts in perceived threat and institutional trust during the vaccine rollout period. As vaccines became more available and public health infrastructure adapted, people likely felt more capable of accessing the vaccine and viewed COVID-19 as somewhat less threatening due to improved treatments and declining case fatality rates [32].*


By comparing individuals’ intention to receive a COVID-19 vaccine before doses were publicly available with their intentions after rollout, this longitudinal study fills critical knowledge gaps about how real-world vaccine availability, evolving risk perceptions, and shifting trust in government shape changes in vaccination decision-making over time.

## 2. Materials and Methods

In July 2020, we conducted a survey of 788 U.S. adults using survey platform Qualtrics to explore the relationships between demographics and psychosocial predictors of intent to get a future COVID-19 vaccine. Participants were recruited from existing Qualtrics panel participants. To be eligible, participants had to be U.S. residents aged 18 or older and be able to read and understand English. Nine months later, in April 2021, the participants from this study who had indicated being willing to fill out a follow-up survey were sent the same survey questions with a few modifications; N = 142 participants completed both study 1 and study 2. Both studies were approved by the Institutional Review Board at Virginia Commonwealth University, a large public research university in the Mid-Atlantic U.S. (approval #HM20019631). Methods and results are reported here for the N = 142 respondents who completed both surveys.

### 2.1. Measures

Demographics. Demographic variables included age, gender, race/ethnicity, and education.

Trust in Government. Trust in the U.S. government’s handling of the COVID-19 pandemic was measured by the Trust in Government Scale, which uses key components of the Trust Determination Model [34]. Using 4-point Likert responses (with higher numbers indicating a higher level of trust), the Trust in Government Scale contains seven items, measure respondents’ attitudes toward openness, honesty, commitment, caring and concern, and competence of the government in addressing the COVID-19 pandemic; the extent to which respondents believed the government’s actions in response to the pandemic were in their personal interest; and how much respondents believed the government would protect them from COVID-19. Cronbach’s alpha for items on the scale was 0.950 in Time 1 and 0.958 in Time 2. The mean of the seven items was calculated.

Health Belief Model (HBM). Participants responded to each of HBM items described below using a seven-point Likert scale that ranged from “strongly disagree” to “strongly agree” except for the question about ease of access to the vaccine in the Self-efficacy domain, which used a six-point Likert scale ranging from “very difficult” to “very easy.” The constructs all were adapted from a study focused on a pandemic flu vaccine by Myers and Goodwin [22].

Perceived severity of COVID-19 was determined using three items (e.g., “Complications of COVID-19 are serious”). Cronbach’s alpha for items on the scale was 0.756 in Time 1 and 0.754 in Time 2. The mean of the three items at each time point was calculated.

Perceived susceptibility to COVID-19 was measured using three items (e.g., “I am worried about the likelihood of getting COVID-19 in the near future”) [22]. Cronbach’s alpha for items on the scale was 0.811 in Time 1 and 0.823 in Time 2. The mean of the three items at each time point was calculated.

Perceived benefits of a COVID-19 vaccine were measured using four items focused on the benefits of a COVID-19 vaccine (e.g., “Vaccination will decrease my chance of getting COVID-19 or its complications”) [22,35]. Cronbach’s alpha for these items was 0.818 in Time 1 and 0.874 in Time 2, and the mean of the four items at each time point was calculated.

Perceived barriers to a COVID-19 vaccine were measured using seven items (e.g., “The development of a COVID-19 vaccine is too rushed to properly test its safety,” and “I am concerned about the side effects of a COVID-19 vaccination”) [22,35]. Cronbach’s alpha for these items was 0.824 in Time 1 and 0.876 in Time 2, and the mean of the seven items at each time point was calculated.

Self-efficacy was measured by three items (e.g., “How certain are you that you could get a future COVID-19 vaccination?” with responses ranging from “very uncertain” to “very certain”) [22]. Cronbach’s alpha for items on the scale was 0.768 in Time 1 and 0.808 in Time 2. The mean of the three items at each time point was calculated.

Cues to action was measured using one item, “Has a physician, healthcare provider, or clinician spoken to you about getting the COVID-19 vaccine,” with answer options “Yes” and “No.” 

COVID-19 vaccine uptake/intent. In the Time 1 survey, intention to get a future COVID-19 vaccine was measured using one item: “Do you intend to get the COVID-19 vaccine when it becomes available?” with responses ranging from “I will definitely not get the COVID-19 vaccine” to “I will definitely get the COVID-19 vaccine” on a five-point scale. In the Time 2 survey, COVID-19 vaccine uptake/intent was also measured using one item: “Several COVID-19 vaccines have been approved as of early 2021. Have you or are you planning to get the COVID-19 vaccine?” with responses ranging from “I am not planning to get the COVID-19 vaccine” to “I already have gotten the COVID-19 vaccine” on a four-point scale.

Both outcomes were collapsed into a discrete “vaccinated (intent or behavior)/not vaccinated” variable, with “I will definitely get the COVID-19 vaccine” and “I will probably get the COVID-19 vaccine” classified as “Intending to vaccinate” and “ “I am uncertain about whether I will get the COVID-19 vaccine,” “I will probably not get the COVID-19 vaccine,” and “I will not get the COVID-19 vaccine” as “Not intending to vaccinate” in Time 1 and “I already have gotten the COVID-19 vaccine,” “I am planning to get the COVID-19 vaccine as soon as possible” classified as “Vaccinated,” “I am uncertain about whether I will get the COVID-19 vaccine,” and “I will not get the COVID-19 vaccine” as “Unvaccinated” in Time 2. This was done because of the relatively small and well-established intention-behavior gap and strong intention-behavior link for vaccinations [36].

### 2.2. Statistical Approach

Analyses were performed using SPSS 30.0. For both time points, the following statistical analyses were carried out: Differences in race/ethnicity, gender, and education were explored using chi-square tests. Age differences were explored using a *t*-test. Hierarchical multiple logistic regression analyses were used to explore which variables predicted vaccine intention prior to release as well as uptake/intent after release. Demographic variables were entered in Block 1, Trust in Government was entered in Block 2, and Health Belief Model variables were entered in Block 3. The effects of the independent variables at each step were expressed as odds ratios (ORs) with 95% confidence intervals. Model fit and the proportion of variance explained were assessed using the Nagelkerke *R*^2^ at each step. In addition, *t*-tests and a chi-square test were conducted to determine differences in HBM constructs and trust in government between Time 1 and Time 2. No missing data were present in the dataset.

To evaluate model robustness given the modest sample size and number of predictors, we conducted collinearity diagnostics and sensitivity checks. Although several HBM constructs are related within the broader theoretical model, they represent distinct psychological domains—perceived susceptibility, severity, benefits, barriers, and self-efficacy—that are defined independently in the Health Belief Model framework. Consistent with this theoretical separation, variance inflation factors (VIFs) for all predictors were below 2.0, indicating no problematic multicollinearity among HBM constructs, trust in government, or demographic variables. Tolerance statistics also exceeded recommended thresholds (>0.40). As a sensitivity analysis, we reran each hierarchical model using a reduced set of predictors (retaining only those significant at the bivariate level), and the pattern of significant predictors remained unchanged. Together, these diagnostics support the stability of the reported logistic regression findings despite a lower events-per-variable ratio.

## 3. Results

Of 142 respondents who answered both surveys 1 and 2, 42.3% (*n* = 60) were female and 57.7% male (*n* = 82); 33.8% (*n* = 48) White and 66.2% (*n* = 94) persons of color. The mean age of participants was 53.3 (SD: 15.6). Of the sample, 48.6% (*n* = 69) reported having a bachelor’s degree or higher, while 51.4% (*n* = 73) did not. The mean age for those who only answered at Time 2 was significantly higher than for those who answered at both timepoints (*p* < 0.001). Significantly more respondents were female compared to male (*p* = 0.007). There was not a significant difference for race (*p* = 0.758) and level of education attained (*p* = 0.894).

At Time 1, 41.5% of respondents were definitely planning, 26.1% were probably planning, 15.5% were uncertain, 6.3% probably not planning, and 10.6% definitely not planning to get the COVID-19 vaccine; collapsed, 67.6% were considered “Intending to vaccinate” and 32.4% were considered “Not intending to vaccinate.” At Time 2, 31.0% of respondents reported being vaccinated, 38.0% were planning, 11.3% were undecided, and 19.7% were not willing to get a COVID-19 vaccine; collapsed, 69.0% were considered “Intend to vaccinate/vaccinated” and 31.0% were considered “Not intending to vaccinate/not vaccinated” (See Table 1).

### 3.1. Bivariate Analyses

Prior to release of the first COVID-19 vaccines, chi square tests indicated that women (compared to men, *p* = 0.046) were significantly more likely to express intent to get a future COVID-19 vaccination. A *t*-test indicated that older respondents were more likely compared to younger respondents to express intent to get a future COVID-19 vaccination (*p* = 0.027). Education and race/ethnicity were not significant predictors at this time point.

After release of the first COVID-19 vaccines, chi square tests indicated that women (compared to men, *p* = 0.043) were significantly more likely to report COVID-19 uptake/intent. A *t*-test indicated that older respondents were more likely compared to younger respondents to report COVID-19 vaccine uptake/intent (*p* < 0.001). As with Time 1, education and race/ethnicity were not significant predictors.

### 3.2. Psychosocial Predictors of Vaccine Intention/Uptake

Time 1

To investigate future COVID-19 vaccine uptake/intent, prior to the release of the COVID-19 vaccines, a hierarchical logistic regression was carried out to compare the effects of demographics and HBM variables on COVID-19 vaccine uptake/intent. For the first step, demographic variables were entered. Step 1 for COVID-19 vaccine uptake/intent was statistically significant, X^2^(4) = 9.568, *p* = 0.048; however, none of the demographic variables were significant predictors (Table 2). This model explained 9.1% (Nagelkerke *R*^2^) of the variance in COVID-19 vaccine uptake/intent.

For Step 2, trust in government was added as a predictor. Step 2 for COVID-19 vaccine uptake/intent was not statistically significant, X^2^(5) = 10.950, *p* = 0.052. Demographic variables and trust in government were not significant. The Nagelkerke *R*^2^ value of 0.104 suggested that this step accounted for 10.4% of the variance in COVID-19 vaccine uptake/intent.

For Step 3, the HBM variables were added as predictors. Step 3 for COVID-19 vaccine uptake/intent was also statistically significant, X^2^(11) = 92.682, *p* < 0.001. Demographic variables were not significant, and neither was trust in government. Higher perceived susceptibility was associated with higher COVID-19 vaccine uptake/intent (OR: 1.685; *p* = 0.038), higher perceived benefits of the COVID-19 vaccine were associated with higher COVID-19 vaccine uptake/intent (OR: 3.124; *p* < 0.001), and higher perceived barriers to getting the COVID-19 vaccine were associated with lower COVID-19 vaccine uptake/intent (OR: 0.212; *p* = 0.002) (see Table 2). The Nagelkerke *R*^2^ value of 0.669 suggested that this model accounted for 66.9% of the variance in COVID-19 vaccine uptake/intent, and that adding HBM variables accounted for an increase of 56.5% in *R*^2^.

Time 2

To investigate COVID-19 vaccine uptake/intent after the release of the COVID-19 vaccines, a hierarchical logistic regression was carried out to examine the effects of demographics, trust in government, and HBM variables on COVID-19 vaccine uptake/intent. For the first step, demographic variables were entered. Step 1 for COVID-19 vaccine uptake/intent was statistically significant, X^2^(4) = 17.921, *p* = 0.001. Younger respondents were more likely than older respondents to indicate COVID-19 vaccine uptake/intent (OR: 1.041; *p* = 0.002). The other demographic variables were not significant (Table 3). This model explained 16.7% (Nagelkerke *R*^2^) of the variance in COVID-19 vaccine uptake/intent.

Step 2 adding trust in government was as a predictor was statistically significant, X^2^(5) = 54.341, *p* < 0.001. Demographic variables were not significant. Trust in government was associated with higher COVID-19 vaccine uptake/intent (OR: 5.281; *p* < 0.001). The Nagelkerke *R*^2^ value of 0.448 suggested that demographics and trust in government accounted for 44.8% of the variance in COVID-19 vaccine uptake/intent.

Step 3 adding the HBM variables was also statistically significant, X^2^(11) = 115.335, *p* < 0.001. Demographic variables were not significant. Trust in government remained associated with COVID-19 vaccine uptake/intent (OR: 2.817; *p* = 0.047). In contrast to Time 1, perceived susceptibility was no longer significantly associated with COVID-19 vaccine uptake/intent. Higher perceived benefits of the COVID-19 vaccine were still associated with higher COVID-19 vaccine uptake/intent (OR: 3.274; *p* < 0.001), and higher perceived barriers to getting the COVID-19 vaccine were also still associated with lower COVID-19 vaccine uptake/intent (OR: 0.244; *p* = 0.002) (see Table 3). The Nagelkerke *R*^2^ value of 0.782 suggested that this model accounted for 78.2% of the variance in COVID-19 vaccine uptake/intent, and that adding HBM variables accounted for an increase of 33.4% in *R*^2^.

### 3.3. Differences in HBM Constructs and Trust in Government over Time

Finally, a series of T-tests and one chi-square test were carried out to compare the scores for HBM constructs as well as trust in government between Time 1 and Time 2. T-tests showed that perceived severity of COVID-19 decreased significantly between July 2020 and April 2021 (t = 5.186, *p* < 0.001); perceived barriers to the COVID-19 vaccine also decreased (t = 4.518, *p* < 0.001); and self-efficacy to get a COVID-19 vaccine increased (t = −3.477, *p* < 0.001). There were no significant differences between perceived susceptibility in Time 1 and 2, nor between perceived benefits of the COVID-19 vaccine between times 1 and 2 (see Table 4). Finally, trust in government increased significantly between times 1 and 2, t = −8.193, *p* < 0.001.

## 4. Discussion

This study examined predictors of COVID-19 vaccine uptake/intentions and behaviors over time, with data collected before and after the release of the first COVID-19 vaccines. All three hypotheses proposed were supported. First, perceived benefits and perceived barriers emerged as the most robust predictors of vaccine intent and uptake at both time points, but their predictive power became more central after vaccine rollout, supporting the hypothesis that these constructs gain salience as vaccination becomes a tangible choice. Second, trust in government was not significantly associated with vaccine intent prior to vaccine availability but became a significant predictor after vaccines were broadly accessible, aligning with the hypothesis that trust plays a more influential role once institutions are actively implementing health interventions. Finally, consistent with expectations, perceived severity of COVID-19 declined over time, while self-efficacy to obtain a vaccine and trust in government both increased, reflecting evolving public perceptions as the pandemic response matured, and supporting the hypothesis that this construct would decrease over time. The findings provide insight into how sociodemographic and psychosocial factors, particularly Health Belief Model (HBM) constructs, and trust in government, was associated with vaccine-related decision-making during the early phases of the pandemic. However, it is important to note that while temporal associations are shown, causality cannot be inferred.

Consistent with prior research on vaccine uptake during pandemics [37,38], we found that several HBM constructs, particularly perceived benefits and perceived barriers, were strong predictors of COVID-19 vaccine intentions and behaviors. Before vaccine availability, individuals who perceived themselves as more susceptible to COVID-19 and who recognized more benefits and fewer barriers to vaccination were significantly more likely to report intent to vaccinate. After the vaccine became available, perceived susceptibility was no longer a significant predictor, but perceived benefits and barriers remained influential. This pattern suggests that once vaccines were accessible, beliefs about vaccine utility and obstacles to access played a more central role in shaping uptake decisions than beliefs about personal risk.

Interestingly, although age and gender emerged as significant predictors of vaccine intent in bivariate analyses, these effects did not hold in multivariate models that included HBM constructs. This aligns with previous studies showing that psychosocial variables mediate the influence of demographic characteristics on vaccine-related choices. For example, prior research has shown that women often report greater vaccine hesitancy, but may also have stronger health-protective attitudes [39,40], which could explain their greater uptake/intentions here. Similarly, although older adults were more likely to express intent to vaccinate in bivariate analyses, this association disappeared in the presence of HBM predictors, suggesting that age-related differences may be explained by differing health beliefs.

Findings also underscore the importance of conceptualizing trust in government as a dynamic, context-dependent factor in shaping public health behavior. The lack of a significant relationship between trust in government and vaccine uptake/intent at Time 1, contrasted with the significant association at Time 2, aligns with models such as Covello’s Trust Determination Theory [33], which emphasize that trust is continually evaluated based on perceptions of government competence, honesty, consistency, and care. As trust is influenced by both the content and delivery of communication [41], it is likely that public evaluations of governmental performance evolved during the pandemic in response to ongoing events, communication strategies, and vaccine rollout efforts. This sensitivity to timing underscores findings from recent studies showing that changes in institutional trust can have a substantial impact on health behaviors such as vaccine decision-making [42,43]. For public health practitioners and policymakers, the implication is clear: trust-building must be sustained throughout a health crisis, with deliberate attention to transparent communication, responsiveness to public concerns, proactively monitoring trust metrics during outbreaks, and visible modeling of recommended behaviors. These strategies are not only important in the early stages of a pandemic but are crucial over time, particularly when public support is needed for interventions such as vaccination.

Over time, we observed several shifts in measured constructs for this study. Notably, trust in government increased significantly when comparing Time 1 and Time 2. This confirms what has been shown in some studies [44,45], but not in others [46]. For example, one study found decreasing trust in U.S. government health agencies from February 2020 through May 2020 and June 2022 [47]. The increase in trust in government observed in this study between July 2020 and April 2021 may reflect both structural and perceptual shifts distinct to that specific phase of the U.S. pandemic response. This period included a presidential transition, broader vaccine availability, and more consistent public health messaging, all factors that may have enhanced perceptions of government competence and responsiveness [48]. This rise in trust was not just a background shift; it became a significant predictor of vaccine uptake/intentions at Time 2, suggesting that trust in public institutions may act as a lever for promoting uptake once implementation efforts are underway. These findings underscore that trust is not static and can be rebuilt over the course of a public health crisis. For future pandemic planning, this finding highlights the importance of sustained, transparent communication and the potential value of visible leadership transitions or policy resets that realign public perception and reinforce institutional credibility. Efforts to restore trust, even mid-crisis, may translate into downstream benefits in public cooperation with health interventions. Further, trust needs to be cultivated and maintained to ensure that it does not erode again following transitional periods.

In addition, participants reported lower perceived severity of COVID-19 and fewer barriers to vaccination in 2021 compared to 2020, alongside increased self-efficacy. These changes align with broader trends in public understanding and infrastructure improvements. As the pandemic evolved and vaccination campaigns expanded, perceived risk of severe illness may have declined due to better treatments and lower case fatality rates [49], while improved vaccine availability and streamlined processes likely reduced perceived logistical and psychological barriers [50]. Increased self-efficacy may have reflected growing familiarity with the healthcare system’s vaccination procedures and decreasing uncertainty about eligibility or safety protocols [51].

### 4.1. Implications

These findings have several important implications for future pandemic preparedness and future pandemic vaccination campaigns, particularly in the light of decreasing vaccine confidence since the COVID-19 pandemic [14,52,53]. First, messaging that emphasizes the benefits of vaccination, including protection for oneself and others, return to normal activities, and contribution to community health, appears to remain effective across timepoints. Public health campaigns should also proactively address perceived barriers to pandemic vaccination, whether structural (e.g., transportation, cost, scheduling) or psychological (e.g., fear of side effects, misinformation). Doing so may be especially critical in later stages of a vaccination campaign when individual risk perception is declining, but barriers may still deter action. Second, the disappearance of perceived susceptibility as a predictor post-COVID-19 vaccine rollout suggests that risk-based messaging may be less effective over time as perceptions of personal vulnerability diminish. Messaging strategies that evolve alongside public perception and emphasize collective benefit and social responsibility may prove more durable in sustaining uptake [54]. Third, with trust in government declining further since the end of the pandemic [36,37,55], the importance of (re)building this trust cannot be overstated in light of pandemic preparedness. As the current findings show, trust became a key predictor of vaccine uptake once vaccines were available, suggesting that future campaigns may falter without public confidence in institutions. Rebuilding trust will require more than accurate messaging such as visible responsiveness, consistent communication, effective community partnerships (e.g., faith-based organizations, community leaders), and efforts to engage trusted messengers at both national and local levels. Trust must be cultivated before the next crisis, not during it. In addition, attention should be given to how these findings may translate to contexts with different governance models, including low- and middle-income countries where regulatory environments and public health infrastructures may differ substantially. Although this study was carried out in the United States, evidence suggests that trust in governmental institutions is a robust predictor of vaccine attitudes across diverse national contexts [38]. Comparative studies have shown that higher institutional and governmental trust is associated with greater vaccine confidence and uptake in Europe, Asia, and other regions, indicating that this relationship is not unique to the U.S. context [14]. Finally, demographic variables alone may offer limited predictive value for vaccine uptake and should not be the primary basis for targeted interventions. Instead, interventions should be tailored to psychological profiles—particularly around benefit/barrier perceptions and self-efficacy—rather than broad demographic categories alone.

### 4.2. Strengths, Limitations and Future Directions

A primary strength of this study is in its longitudinal nature, which provides a window into predictors of COVID-19 vaccine uptake/intent both at two crucial time points during the pandemic: Before and shortly after the COVID-19 vaccines were approved and became available to the public. However, several limitations should also be considered when interpreting these results. First, the data were self-reported and may be vulnerable to social desirability bias [56,57]. Second, although COVID-19 vaccine uptake/intent was measured at both time points, the phrasing of that question was slightly different between the two waves, which limits direct comparison in uptake/intent within study subjects. These differences were introduced to reflect the evolving pandemic context; intent at Time 1 necessarily referenced a hypothetical future vaccine, whereas the Time 2 item assessed actual uptake or ongoing intent once vaccines were available. Although this limits strict longitudinal comparability, the constructs captured at both time points represent the closest conceptual equivalents available in their respective phases, and both map onto the same underlying behavioral outcome (vaccination behavior vs. non-vaccination). Therefore, the items are appropriate for examining shifts in predictors over time, even if direct numerical comparison of intent levels should be interpreted cautiously. Third, the sample may not fully reflect the diversity of the general population, limiting generalizability. While race/ethnicity was not a significant predictor in this study, broader research has documented racial disparities in COVID-19 vaccine access and uptake [58], which merit further exploration. Additionally, the study did not control for political orientation, which has emerged a notable determinant of vaccine perceptions and trust in government institutions [59]. Fourth, although we reported internal consistency estimates for all adapted scales, the study did not include pretesting, cognitive interviewing, or independent validation of these adapted items within the present sample. As a result, measurement error cannot be ruled out, and future research should confirm the psychometric properties of these constructs using more extensive validation procedures.

A further consideration involves the external validity of the findings. Although the demographic characteristics of our longitudinal subsample (e.g., 66% persons of color and nearly half with a bachelor’s degree or higher) are strengths for examining vaccine perceptions among groups that have been historically underrepresented or disproportionately affected by COVID-19, they may also limit generalizability to the broader U.S. population. National demographic distributions differ from those observed in this sample, and vaccination attitudes have been shown to vary across sociodemographic strata. As a result, the associations detected, particularly within multivariate models that include numerous psychosocial predictors, should be interpreted with caution, as the direction or magnitude of effects may differ in more nationally representative samples. Future work using probability-based or weighted sampling approaches will be important for validating these findings and assessing their applicability to diverse U.S. subpopulations.

Fifth, only 142 of the original 788 respondents participated in both surveys, which may have led to some level of attrition bias. Additionally, some of the effects reported, particularly trust in government, were statistically significant but with a wide confidence interval, and should be validated in future studies, particularly considering potentially significant unmeasured confounders such as exposure to misinformation and regional political differences. These wide confidence intervals, especially for trust in government and perceived benefits, likely reflect sampling variability related to the modest sample size and may indicate some degree of model overfitting, underscoring the need for replication with larger and more demographically representative samples. Finally, although our events-per-variable ratio approached the lower recommended threshold for logistic regression, collinearity diagnostics and sensitivity checks indicated that model results were stable and not meaningfully affected by the number of predictors. Nevertheless, even theoretically distinct HBM constructs can show modest empirical association, and future studies may consider structural equation modeling or latent variable approaches to examine how these constructs relate within larger samples. Finally, the second study took place in April 2021, during the prime initial rolling out of the original COVID-19 vaccines, when media attention and policy dynamics were unique, and generalizing to later phases should be done with caution.

## 5. Conclusions

This longitudinal study offers important insights into the psychosocial predictors of COVID-19 vaccine intentions and uptake across two critical phases of the pandemic: before and after vaccine availability. Findings underscore the enduring influence of perceived benefits and barriers on vaccine-related decisions, as well as the increasing role of institutional trust as public health interventions move from planning to implementation. While perceived susceptibility was initially a significant motivator of intent, its influence faded once vaccines became accessible, highlighting the importance of evolving health messaging as public perception shifts.

## Figures and Tables

**Figure 1 vaccines-13-01201-f001:**
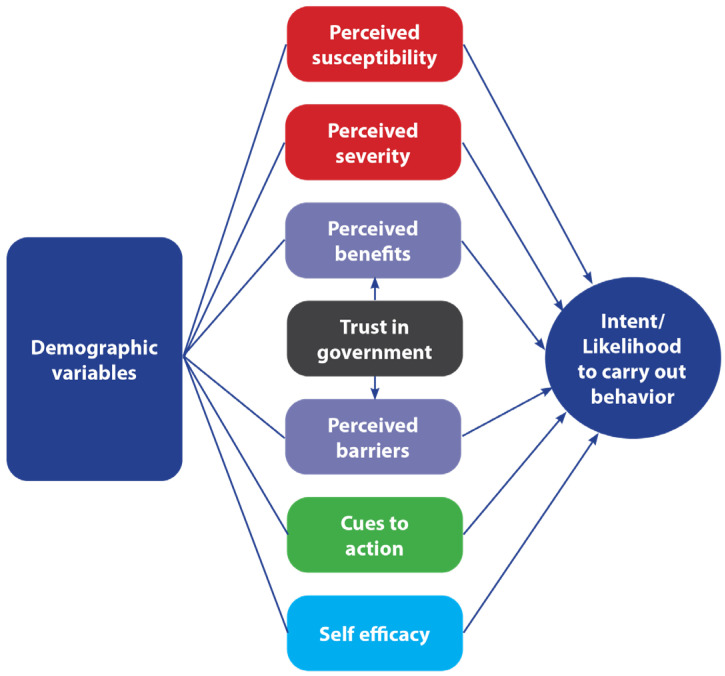
Health Belief Model.

**Table 1 vaccines-13-01201-t001:** Descriptives of the sample (N = 142).

Characteristics	Frequency	*p*-Value Intent Time 1	*p*-Value Uptake Time 2
Gender		0.046 *	0.043 *
Female	42.3% (*n* = 60)		
Male	57.7% (*n* = 82)		
Age, years		0.027 *	<0.001 *
Mean, SD	53.3, 15.64		
Race/ethnicity		0.835	0.864
White	33.8% (*n* = 48)		
Persons of Color	66.2% (*n* = 94)		
Education		0.139	0.118
Less than bachelor’s	51.4% (*n* = 73)		
Bachelor’s or higher	48.6% (*n* = 69)		

* *p* < 0.05.

**Table 2 vaccines-13-01201-t002:** Hierarchical logistic regression predicting COVID-19 vaccine uptake/intent prior to release.

Variable	OR	95% CI	*p*-Value	OR	95% CI	*p*-Value	OR	95% CI	*p*-Value
Age	1.02	0.99, 1.05	0.078	1.02	0.99, 1.06	0.060	1.03	0.99, 1.07	0.188
Gender: Female (Ref: Male)	0.45	1.19, 1.09	0.545	0.49	0.20, 1.20	0.116	0.61	0.14, 2.73	0.522
Race: White (Ref: POC)	0.53	0.20, 1.36	0.185	0.48	0.18, 1.26	0.135	0.66	0.14, 3.10	0.599
Education: Bachelor’s (Ref: Less than bachelor’s)	1.59	0.76, 3.35	0.222	1.55	0.73, 3.29	0.249	1.73	0.53, 5.67	0.365
Trust in government				1.38	0.80, 2.37	0.249	0.87	0.30, 2.57	0.804
HBM: perceived severity							1.64	0.90, 3.00	0.107
HBM: perceived susceptibility							1.69	1.03, 2.76	0.038 *
HBM: perceived benefits							3.12	1.62, 6.04	<0.001 *
HBM: perceived barriers							0.21	0.08, 0.56	0.002 *
HBM: self-efficacy							1.41	0.90, 2.20	0.134
HBM: cues to action							3.13	0.88, 11.15	0.078

* *p* < 0.05. Note: HBM: Health Belief Model; OR: Odd Ratio; CI: Confidence Interval.

**Table 3 vaccines-13-01201-t003:** Hierarchical logistic regression predicting COVID-19 vaccine uptake/intent after release.

Variable	OR	95% CI	*p*-Value	OR	95% CI	*p*-Value	OR	95% CI	*p*-Value
Age	1.04	1.02, 1.07	0.002 *	1.02	0.99, 1.06	0.165	0.99	0.95, 1.04	0.688
Gender: Female (Ref: Male)	0.41	0.16, 1.04	0.060	0.35	0.11, 1.05	0.061	0.21	0.35, 1.25	0.085
Race: White (Ref: POC)	0.37	0.13, 1.01	0.052	0.64	0.19, 2.13	0.469	0.69	0.10, 4.49	0.693
Education: Bachelor’s (Ref: Less than bachelor’s)	1.72	0.79, 3.74	0.170	1.21	0.48, 3.07	0.691	1.69	0.37, 7.85	0.501
Trust in government				5.18	2.73, 9.86	<0.001 *	2.82	1.01, 7.83	0.047 *
HBM: perceived severity							1.22	0.64, 2.34	0.540
HBM: perceived susceptibility							1.51	0.81, 2.82	0.200
HBM: perceived benefits							3.27	1.71, 6.26	<0.001 *
HBM: perceived barriers							0.24	0.10, 0.59	0.002 *
HBM: self-efficacy							0.67	0.42, 1.08	0.098
HBM: cues to action							3.46	0.75, 16.20	0.116

* *p* < 0.05. Note: HBM: Health Belief Model; OR: Odd Ratio; CI: Confidence Interval.

**Table 4 vaccines-13-01201-t004:** Change in HBM constructs over time.

HBM Construct	Time 1	Time 2	*p*-Value
Perceived severity	5.5, 1.33	4.5, 1.68	<0.001 *
Perceived susceptibility	4.5, 1.57	4.2, 1.62	0.067
Perceived benefits	4.6, 1.30	4.6, 1.50	0.864
Perceived barriers	3.7, 1.00	3.1, 1.19	<0.001 *
Self-efficacy	5.0, 1.51	5.6, 1.5	<0.001 *
Cues to action			1.000
Yes	*n* = 92	*n* = 92	
No	*n* = 50	*n* = 50	

* *p* < 0.05. Note: HBM: Health Belief Model.

## Data Availability

The dataset for this study is available from the corresponding author upon request.

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
