# Peer review of "Building Vaccine Readiness for Future Pandemics: Insights from COVID-19 Vaccine Intent and Uptake"

_vaccines, 2025, doi:10.3390/vaccines13121201_

Round 1

Reviewer 1 Report

Comments and Suggestions for Authors

Dear Authors,

The manuscript is well-written and methodologically sound. It contributes to understanding the psychosocial determinants of vaccine intent and uptake within the Health Belief Model and adds longitudinal data necessary for the pandemic preparedness literature. However, several areas require clarification and tightening before publication.

  1. The introduction could be slightly condensed by trimming early sections summarizing the general COVID-19 vaccine development history.
  2. The small sample size limits generalizability and statistical power, particularly in multivariate models with numerous predictors. The demographic distribution (66% persons of color, 48.6% with bachelor’s degree or higher) may not reflect U.S. national demographics, which could bias interpretation. The authors mentioned this, but the implications deserve more explicit discussion regarding external validity.
  3. Hierarchical logistic regression is appropriate. However, the ratio of predictors to total cases may approach the lower threshold for stability in logistic regression (recommended minimum ~10 events per predictor). The authors should provide a sensitivity or collinearity assessment to ensure robustness.
  4. Confidence intervals for key predictors (trust in government and perceived benefits) are wide, suggesting possible overfitting or sampling variability. This should be acknowledged in the Limitations section.
  5. The Time 1 and Time 2 survey items for vaccine intent differ slightly in wording and response scale. Although noted, this difference may have affected longitudinal comparability; a justification should be given.
  6. The finding that trust in government became significant only after vaccine availability supports the second hypothesis. However, the narrative interpretation may slightly overstate causality. The manuscript should emphasize that while temporal associations are shown, causation cannot be inferred due to potential unmeasured confounders (e.g., political ideology, exposure to misinformation, regional policy differences).
  7. The policy relevance section could be expanded to include how governments might proactively monitor trust metrics during pandemics, or the role of partnerships with non-governmental community organizations to enhance vaccine confidence.
  8. In Tables:
    • Ensure consistent decimal precision (e.g., ORs to two decimal places).
    • Ensure consistent labeling (Person of color or Persons of color or Non-White respondents).
    • Verify that abbreviations (HBM, OR, CI) are defined in each table’s caption.
  9. Consider adding a conceptual diagram of the HBM constructs and their hypothesised influence on intent/uptake.
  10. The authors mention social desirability bias, sampling, and the lack of control for political orientation. Two additional limitations should be added:

    • Attrition bias: Only 142 of 788 original respondents completed both surveys, possibly skewing results toward more compliant or health-motivated individuals.

    • Temporal context: The April 2021 follow-up occurred during the U.S. vaccine rollout peak, when media attention and policy dynamics were unique. Generalizing to later phases may not hold.

  11. Policy implications are sound, but could be expanded:

    • Suggest including practical approaches to rebuild trust (community partnerships, transparent communication).

    • Address how these results may apply to low- and middle-income countries or different governance contexts.

  12. Add a few recent post-pandemic references (2023–2025) on evolving vaccine hesitancy trends and trust decline to ensure currency.

Reviewer 2 Report

Comments and Suggestions for Authors

Thank you for the opportunity to review this manuscript. This manuscript addresses an important and timely topic: psychosocial predictors of COVID-19 vaccine intent and uptake, with implications for future events. The longitudinal design (two spaced assessments) is a good approach to document change. Use of the Health Belief Model (HBM) and trust in government are helpful to provide theoretical grounding. The paper is well organized and generally clear, but several sections would benefit from tightening, clarification, and additional information.

Background

  • The background section is overly detailed and could be condensed. 1.1 and 1.2 include extended historical and technical information about vaccine development timelines that are not fully necessary.

  • Consider consolidating sections 1.2 and 1.3 to be more concise and focused on the key problem(s) you are addressing: predictors of vaccine intent and uptake, particularly the role of trust in government.

  • The introduction would be strengthened by more explicitly addressing the knowledge gap this study fills, namely the longitudinal nature.

Theoretical Framework

  • Love the inclusion of the HBM and Trust Determination Theory. Consider adding a conceptual figure to demonstrate this in relation to your hypotheses. That may be helpful for this section as well as for your discussion section.

Methods

  • I appreciate that you reported Cronbach's alphas in this study. Consider describing any pretesting or other validation measures, since you adapted assessments with validity evidence. If you didn't do this, please add as a limitation.
  • Consider addressing the attrition rate more thoroughly and whether or not those that did not complete both assessments were different. It could be done in writing or with a table comparing demographic or other baseline information.

Results

  • Love all of the data in the tables! Consider bolding key information to improve readability.
  • Did you check for multicollinearity across HBM constructions OR consider running more advanced modeling, given the conceptual overlap? (Methods/Results comment)

Discussion

  • Consider organizing and aligning this section more explicitly and clearly with the HBM and Trust Determination Theories. It is interspersed and definitely present, but this could be a good organization for this section.
  • Consider including a short discussion whether trust in government is US-based or broader than that. Maybe cite a reference or two, given the international readership of the journal.
  • Consider concision in the policy implications section.
  • Address additional limitations, as noted above related to attrition, survey pretesting, etc.

I hope this is helpful! :)

Round 2

Reviewer 2 Report

Comments and Suggestions for Authors

Thank you so much for your careful revisions. You have done an excellent job (love the figure), and I have no further comments.